



# Associations of interannual variation of Summer Tropospheric Ozone with Western Pacific Subtropical High in China from 1999 to 2017

**Authors**:Xiaodong Zhang[1,#], Ruiyu Zhugu[1,#], Xiaohu Jian[1], Xinrui Liu[1], Kaijie Chen[1], Shu Tao[1], Junfeng Liu[1], Hong Gao[2], Tao Huang[2], Jianmin Ma[1,*]

**Affiliations:**

[1]Laboratory for Earth Surface Processes, College of Urban and Environmental Sciences, Peking University, Beijing 100871, China

[2]Key Laboratory for Environmental Pollution Prediction and Control, College of Earth and Environmental Sciences, Lanzhou University, Lanzhou 730000, China

[#] These authors contribute equally to this article

[*] Corresponding author. E-mail address: jmma@pku.edu.cn (J.M. Ma).

## Abstract

Associations between tropospheric ozone ($O_3$) and climate variations have been extensively investigated worldwide. However, given the lack of historical $O_3$ monitoring data, the knowledge gaps regarding the influences of climate variations on long-term $O_3$ trends in China remain. The present study used a unique tropospheric $O_3$ dataset from the summer of 1999 to 2017 simulated by an atmospheric chemistry model to explore the linkage between summer $O_3$ and a dominant atmospheric circulation system – the Western Pacific Subtropical High Pressure (WPSH) on an interannual basis in China. During this period, both WPSH strength and $O_3$ concentrations in eastern and central China illustrated a growing trend. An EOF analysis was conducted to examine significant summer $O_3$ characteristics and patterns and their potential connections with the WPSH. We show that the WPSH determines interannual fluctuations of summer $O_3$, whereas $O_3$ precursor emissions contribute primarily to the $O_3$ long-term trend. Special efforts were made to discern the associations of $O_3$ variations in major urban agglomerations of China and the WPSH. The results reveal that the WPSH plays a more vital role in $O_3$ perturbation in the eastern seaboard regions





and inland China, but leads to lower $O_3$ levels in the Pearl River Delta (PRD) region.
Precursor emissions made more significant contributions up to 60% to increasing $O_3$
trends in the inland urban agglomerations than coastal regions in eastern and southern
China. The strongest contribution of meteorological conditions associated with the
WPSH to summer ozone concentration occurred in the Yangtze River Delta (YRD),
accounting for over 9% to ozone perturbations from 1999 to 2017. Overall, we find that
the effect of the WPSH on regional $O_3$ depends on the spatial proximity to the WPSH.
We attributed the effects of the WPSH on $O_3$ interannual variations to the changes in
air temperature, precipitation, and winds associated with the WPSH's intensity and
positions.
**Keywords:** tropospheric ozone, western pacific subtropical high, climate, EOF analysis

**1. Introduction**

Tropospheric (or surface) ozone is one of the most important components of

atmospheric chemistry and is also a prominent atmospheric pollutant in China in recent
years (Ma et al., 2021). Ground-level ozone pollution has overtaken $PM_{2.5}$ as the
leading pollutant in many of China's urban and industrial regions (Lu et al., 2018).
Surface ozone is produced through the photochemical oxidation of carbon monoxide
(CO) and volatile organic compounds (VOCs) in the presence of nitrogen oxides ($NO_x$)
and sunlight (Akimoto et al., 2015; Liu and Wang, 2020; Lu et al., 2018; Ma et al.,
2021). Unlike stratospheric ozone, which absorbs harmful UV radiation that could
otherwise reach the Earth's surface and cause adverse health impacts on humans,
surface ozone has detrimental effects on both human health and terrestrial vegetation
(Fleming et al., 2018; Lefohn et al., 2017; Liu et al., 2018; Liu and Wang, 2020).
Extensive studies have revealed significant associations between short-term or acute
exposure to ozone concentrations and respiratory and cardiovascular morbidity,
inhibiting lung development, new onset asthma, hospital admissions, and premature
mortality (Bell et al., 2014; Fleming et al., 2018; Yan et al., 2013). It is estimated that





death related to ozone exposure comprises 5–20% of all those caused by air pollution
(Brauer et al., 2012; Monks et al., 2015; Silva et al., 2013). In the past decade, partly
due to rapid economic growth and urbanization in China, surface $O_3$ has increased
dramatically (Maji et al., 2019; Zhan et al., 2018). Many urban areas across China have
experienced growing ozone pollution, despite implementing various stringent emission
reduction measures since 2013 (Bell et al., 2014; Liu and Wang, 2020; Yan et al., 2013).
Although the median ozone values exhibit no significant disparity between China and
many industrialized countries and regions such as Japan, South Korea, Europe, and the
United States (US), the frequency of high-ozone events in China is much higher than
those developed countries and regions (Lu et al., 2018; Ma et al., 2016; Xu et al., 2016).

Surface ozone formation and evolution rely on meteorology, atmospheric

chemistry, and the emissions of $O_3$ precursors, such as VOCs and $NO_x$ emitted from
fuel combustion (Li et al., 2020; Ma et al., 2021). Meteorological parameters affecting
surface $O_3$ evolution include but are not limited to winds, air temperature, relative
humidity, and solar radiation (Ma et al., 2021). While anthropogenic factors play vital
roles in ozone formation, meteorological factors determine, to a significant extent, the
changes and evolution in $O_3$ concentrations (Ding et al., 2019; Li et al., 2019, 2020; Lin
et al., 2021, 2022). Meteorological conditions modulate $O_3$ concentrations through
atmospheric transport and affect natural emissions from biological sources and
chemical reaction rates (Fu et al., 2019; Li et al., 2020; Lu et al., 2019). Extensive
investigations have been devoted to short-term, such as hourly and diurnal changes in
$O_3$ levels and their associations with meteorological conditions (Dang et al., 2021; Han
et al., 2020). Given the strong connections between $O_3$ concentration and air
temperature, atmospheric humidity, and winds, interannual and longer-term variations
of $O_3$ are also elucidated in China and worldwide (Chen et al., 2020; Li et al., 2020).
Daily and interannual variations of summertime surface $O_3$ have been linked with
atmospheric teleconnection patterns, such as the ENSO (El Niño-Southern Oscillation),
East Asian summer monsoon, and the WPSH (Liu et al., 2019a; Wang et al., 2016;
Yang et al., 2022; Yihui and Chan, 2005; Yin et al., 2019; Zhao and Wang, 2017; Zhou
et al., 2009). These climate teleconnection patterns provide dynamic and



thermodynamic backgrounds of regional and large-scale weather systems that could
markedly affect the atmospheric pressure, temperature, and winds. Using modeled $O_3$
time series across China from 1999 to 2017, we have examined the response of gridded
summer $O_3$ concentrations to the East Asian Summer Monsoon Index (EASMI), Nino
indices, and western North Pacific subtropical high index (WPSH-I) on an annual basis
in the six major urban agglomerations in China (UAs, Zhang, et al., 2022). The results
revealed that interannual changes in summer $O_3$ in these UAs were more significantly
associated with the WPSH-I among these atmospheric teleconnection patterns. The
finding motivates us to carry out more broad and deep investigations of the associations
between the long-term change in summer $O_3$ and the WPSH, aiming to shed new light
on the extent of the impact of climate variation on $O_3$ trends in urban China.

Limited studies have been carried out to examine the linkage of summer $O_3$ in

China with the WPSH (Jiang et al., 2021; Yin et al., 2019; Zhao and Wang, 2017; Liu
et al., 2019a). These studies all focused on the response of daily summer $O_3$ variation
to the WPSH in eastern China using measured $O_3$ concentrations within a short period
(e.g., 2015-2018, Yin et al., 2019) rather than interannual or longer ozone trends in
mainland China. To fill this knowledge gap, we performed multiple atmospheric
chemistry model simulations of summer (June, July, and August) $O_3$ concentrations
across China from 1999 to 2017. This unique $O_3$ dataset enables us to explore the
responses of the long-term trend and interannual variation of $O_3$ concentrations to
climate variations and to take a broader look at the associations between ozone
evolution and the Western Pacific subtropical high in China (Zhang et al., 2022).
**2. Methodology**
**2.1. WRF-Chem Model Configuration**

The Weather Research and Forecasting model coupled with Chemistry (WRF-

Chem) v3.7 (http://www2.mmm.ucar.edu/wrf/users/wrf_files/wrfv3.7/updates-
3.7.html) was employed to quantify the influences of the WPSH on $O_3$ variation in
China. The model covers mainland China with a 20 km × 20 km grid resolution,



extending from the ground surface to 50 hPa with 30 non-uniformly distributed verticle
layers (Zhang et al., 2022). Anthropogenic emissions data of atmospheric pollutants
from 1998 to 2017 were collected from EDGAR (Emissions Database for Global
Atmospheric Research) v4.3 (https://edgar.jrc.ec.europa.eu/), including gridded annual
emission data for $CH_4$, BC, OC, $NH_3$, NMVOC, $NO_x$, CO, $SO_2$, and primary $PM_{10}$ and
$PM_{2.5}$. The biogenic emissions were estimated by the MEGAN v2.1 (Model of
Emissions of Gases and Aerosols from Nature) (Guenther et al., 2012). Detailed WRF-
Chem configuration, modeling setup, and precursor emissions are referred to by Zhang
et al. (2022). WRF-Chem model was integrated to predict daily $O_3$ concentrations in
summer (June to August) from 1998 to 2017. After excluding the model spin-up time,
the $O_3$ time series from 1999 to 2017 was used in the present study. The daily
concentrations were summed and averaged over the summer season to obtain mean $O_3$
concentrations. The modeled $O_3$ concentrations were verified by measured $O_3$
concentration data in several major urban agglomerations across China. More details
are referred to in Supporting Information Text 1 and **Fig. S1**.
**2.2. WPSH index**

The WPSH indices were collected from the National Climate Center of China

(NCCC,    the    WPSH    index    is    available    at    http://cmdp.ncc-
cma.net/download/precipitation/diagnosis/NWP_high/wpsh_idx.txt).    The    NCCC
reports four WPSH indices, including the WPSH area index, intensity index, the
westernmost point, and the ridgeline index of the WPSH. These indices define and
quantify the changes in the WPSH via its size, intensity, east–west expansion, and
north–south movement (Liu et al., 2019b). These WPSH activities significantly affect
China's daily, seasonal, interannual, and longer-term meteorological fields and climate
variations. Among the four WPSH indices, we found that the WPSH area index
(hereafter referred to as WPSH-I1) exhibited the most significant positive correlations
with modeled summer ozone concentrations in most regions of China. The strongest
negative correlations occur between $O_3$ concentrations and the westernmost point of the





WPSH (hereafter referred to as WPSH-I2). In light of this, we chose the WPSH-I1 and
WPSH-I2 to elucidate the potential influences of the WPSH on the interannual
variations of WRF-Chem simulated summer $O_3$ concentrations for the past two decades.
As shown in **Fig. S2**, the WPSH strength characterized by the WPSH-I1 index
illustrates a growing trend after 1999, suggesting the reinforcement of the WPSH on a
decadal scale in the recent two decades, the period coincident with the most rapidly
growing $O_3$ pollution in China. This trend possibly overwhelms interannual changes in
the WPSH in the recent two decades.
**2.3. $O_3$ data**
Surface $O_3$ concentration data on a daily basis used the WRF-Chem simulated
concentration data (section 2.1). Meteorological data used the WRF predicted gridded
air temperature (C°), 500-hPa geopotential height (GH, ghm), winds, and the sea
surface pressure (SSP, hPa). To perform the composite analysis for examining the
responses of interannual variation of summer ozone to the WPSH, we also collected
geopotential height at the 500 hPa, the surface air temperature (°C), and precipitation
from NCEP reanalysis (https://psl.noaa.gov/data/reanalysis/reanalysis.shtml). These
data were used to illustrate the characteristics of meteorological fields during the
positive and negative phases of WPSH indices and in the first EOF loadings, which will
be elaborated on below.
**2.4. EOF analysis**
To extract the potential influences of the interannual changes in the WPSH on $O_3$
variations, we conducted the EOF analysis and examined associations between
meteorological fields and surface $O_3$ from 1999 to 2017, respectively. The empirical
orthogonal function (EOF) analysis as a multivariate statistical technique has been
extensively used in atmospheric science to explore the spatiotemporal variations in a
meteorological variable or air pollutant (Fiore et al., 2003; Pu et al., 2016; Shen et al.,
2015; Yin et al., 2019; Zhao and Wang, 2017). In the present study, we used the EOF



173 analysis in WRF-Chem simulated gridded (20 km × 20 km) seasonal $O_3$ concentrations

174 across China to extract annual $O_3$ change features from 1999 to 2017, respectively. The

175 EOF analysis of the $O_3$ concentration time series from modeled data was designed to

176 investigate potential associations between the summer $O_3$ time series and WPSH and

177 to explore the response of the $O_3$ time series to increasing WPSH strength since 1999.

178 The orthogonal modes included spatial and temporal coefficients and contained

179 information of some proportion (variance contributions) from the original fields.

**2.5. Model scenario setup**

181  We quantify the contribution of meteorology and precursor emissions to $O_3$

182 evolution subject to WPSH by setting up three model scenarios. Considering the

183 increasing trend of the WPSH from 1999 onward, we integrated WRF-Chem from 1998

184 to 2017, subject to three model runs. The first model scenario run took the variable

185 meteorological field and annual $O_3$ precursor emissions from 1998 to 2017, with 1998

186 as the model spin-up period, referred to as the base scenario (scenario 1); the second

187 scenario run adopted fixed precursor emissions in 1998, but variable meteorology

188 throughout 1998 to 2017, referred to as model scenario 2, and the third scenario

189 implemented fixed meteorology in 1998 but variable precursor emissions, referred to

190 as model scenario 3. The simulated $O_3$ concentrations from these three scenarios were

191 compared to identify the relative significance of meteorology and precursor emissions

192 in the changes in $O_3$ concentrations.

**3. Results and Discussion**

**3.1. EOF analysis**

196  **Figures 1a** and **1b** show modeled summer mean $O_3$ concentrations and standard

197 deviations (STD) averaged from 1999 to 2017. Higher concentrations are observed in

198 Sichuan and the region extending from central China to the Northern China Plain (NCP),

199 rather than the southern and southeastern seaboard areas where $O_3$ pollution has been





receiving extensive concerns (**Fig. 1a**). This spatial distribution pattern agrees well with
measured mean summer concentrations data averaged from 2015 to 2017 in China (**Fig.**
**S3**). The STD distribution does not superimpose with $O_3$ concentrations but is centered
in the Sichuan Basin and those provinces in the middle reaches of the Yangtze River,
implying that $O_3$ fluctuated more strongly by interannual variations of meteorological
fields in this region.

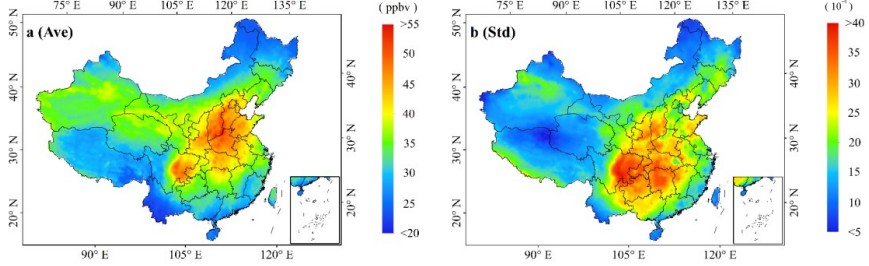

**Figure 1.** Mean summer $O_3$ concentrations (**a**) and standard deviations (**b**) averaged from 1999 to
2017.


We carried out an EOF analysis by using summer $O_3$ as the original field to

illustrate the spatiotemporal variation of $O_3$ in China on an annual basis from 1999 to
2017, aiming to explore the response of summer $O_3$ interannual (1999 to 2017) variation
to the WPSH, the period matching the significantly increasing trend of the WPSH-I1
(Supporting Information (SI), Inset figure of **Fig. S2**), which may lead to a more robust
response of the $O_3$ time series to the WPSH. The results of the first and second EOF
patterns for both periods are presented in **Fig. 2**. Each EOF spatial pattern represents a
share of the total variation of surface ozone proportional to its eigenvalue. The first
EOF loadings (PCA1) are associated strongly with the mean summer $O_3$ concentrations
averaged over the six UAs in China at the correlation coefficient of 0.95 (p<0.01) from
1999 to 2017. The EOF1 pattern also illustrates similarities with the mean summer $O_3$
concentrations and its standard deviations (**Fig. 1a** and **1b**), featured by large values in
central China. Differing from the EOF1, the EOF2 patterns show a south-north contrast
pattern. During this period, the first EOF pattern (EOF1) explains 67.4% of the total
variance in summer $O_3$, and the second EOF pattern (EOF2) explains 9.7% of the total



variance. The negative and positive values in the EOF patterns are expected to represent
the extent of departures from the average summer ozone. Since the EOF1 pattern is the
maximum possible fraction of the variability in the original data, in our case, it explains
most of the summer $O_3$ variability, featured by the growing trends of summer $O_3$
concentrations. The EOF1 pattern appears to agree, to a large extent, with measured
summer (June to August) and warm season (April to September) MDA8 (maximum 8h
average) $O_3$ distribution (Lu et al., 2018; Liu, 2020). The EOF2 patterns also agree with
the second EOF pattern that Yin et al. (2019) obtained, though their EOF analysis
focused on daily $O_3$ in eastern China. The result suggests that the NCP suffered from
higher $O_3$ pollution and was also subject to $O_3$ evolution during the past decades.

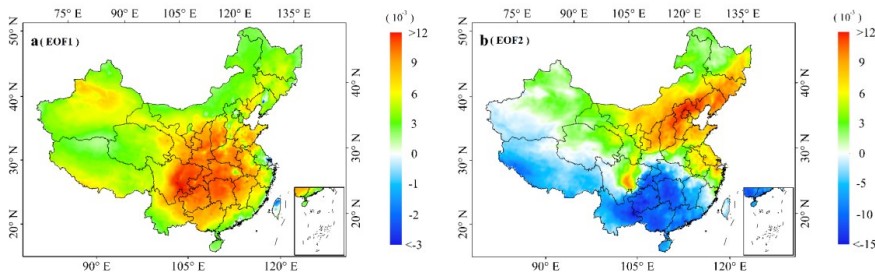


**Figure 2.** First (a) and second (b) EOF patterns across China from 1999 to 2017.

The EOF1 shown in **Figs. 2a** suggests that the most significant variations in

summer $O_3$ occurred in inland areas of China, extending from Sichuan Province to the
middle and lower reaches of the Yangtze River and from Hunan to Shanxi Province.
This inland region covers several major urban agglomerations (UAs) in China,
including Central China (CC), Middle Reaches of the Yangtze River (MYR), and
Chengyu (CY, Chengdu−Chongqing) urban agglomeration (Zhang et al., 2022). We
estimated the correlation coefficients between the first EOF loading (PCA1) and
summer $O_3$ concentrations in the six UAs, where 34.3% of China's population resides.
The results are presented in **Fig. S4**. Strong statistically significant correlations were
found in CC ($r = 0.86$, $p<0.01$), CY ($r = 0.92$, $p<0.01$), and MYR ($r = 0.90$, $p<0.01$).
Whereas, in the other three UAs located near the coastal regions, namely the YRD,



PRD, and BTH, the correlation coefficients range from 0.36 to 0.51 (**Fig. S4**). In
particular, the PCA1 exhibits more strong association with the summer $O_3$ anomalies
averaged over the six UAs, reaching $r$=0.94 (p<0.01). The good correlations between
$O_3$ concentrations and PCA1 are expected because, as aforementioned in section 3.1
that, the EOF analysis was carried out by using summer $O_3$ concentrations as the
original field that have been increasing during the past decades. However, the
magnitude of the correlation coefficients helps identify the extent of $O_3$ pollution and
long-term growth trends in China and different UAs (or regions). Overall, these results
confirm a more substantial interannual variation of summer $O_3$ in the inland areas than
in coastal regions of southern and southeastern China.

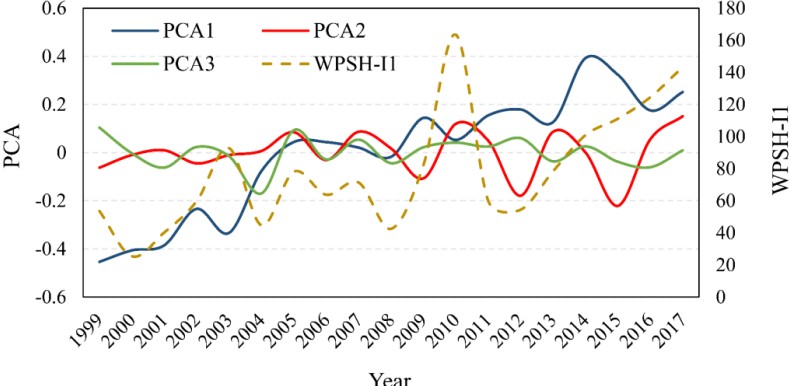

**Figure 3.** Annual variation of three EOF loadings (PCA1-3, scaled on the left Y-axis) and WPSH-
I1 (dashed brown line, scale on the right Y-axis) from 1999 to 2017.

**Figure 3** shows annual variations of the three EOF loadings (PCA1-3), scaled on

the left Y-axis) and the WPSH-I1 (dashed brown line, scaled on the right Y-axis) from
1999 to 2017. The first EOF loading (PCA1) and WPSH-I1 exhibit growing trends
during this period with a correlation coefficient of 0.56 (p<0.01). The increasing trend
of WPSH-I1 since 1999 likely anticipates the interdecadal variation of the WPSH for
the recent two decades. Since $O_3$ concentrations are positively correlated with the
WPSH-I1 (**Figs. 3-5**), stronger WPSH intensity might elevate summer $O_3$ levels in
China on an annual basis, particularly the areas with large EOF1 values in inland China





(**Fig. 2a**). However, this conclusion is not well applicable in the PRD region, where we
observed the lowest association between the EOF1 and summer $O_3$ concentrations (**Fig.**
**2**) and between the WPSH-I1 and $O_3$ levels among the six UAs and over China (**Fig.**
**S4**). It is also worthwhile to note that, because $O_3$ precursor emissions in China have
been growing during the past two decades and modeled concentrations were mainly
attributed to precursor emissions, the positive correlations between $O_3$ concentrations
and the WPSHI-I1 should not be understood that the WPSH drove elevated $O_3$ for a
long term perspective. Further discussions are provided in the next section.
We further compared the 500-hPa geopotential heights (GH, gpm) anomalies in
the positive and negative phases of PCA1 and WPSH-I1 as the departure from their
respective means averaged from 1999 to 2017. We selected those years with the
positive and negative anomalies of the PCA1 and WPSH-I1 $\geq \pm 1$ standard deviation
(STD, referred to as the positive and negative phase hereafter) and then estimated their
composite means. The results are shown in **Figs. S5** and **S6**. It can be seen that the
composite means of 500-hPa geopotential height in the positive and negative phases of
the PCA1 and WPSH-I1 illustrate good spatial similarities, again demonstrating the
connections between summer $O_3$ and WPSH-I1. In the positive phase, positive
geopotential height anomalies at the 500-hPa governed China, except for the NCP
regions, including the BTH urban agglomeration, where negative anomalies of the 500-
hPa geopotential heights are observed. On the other hand, a south-north contrast pattern
of the geopotential height composite anomalies is discerned in the negative phase of
the PCA1 and WPSH-I1. The spatial patterns of GH composite anomalies in the
positive and negative phases of the EOF1 also exhibit some similarities with the GH
composite anomalies based on positive and negative $O_3$ concentration anomalies as the
departure from mean $O_3$ levels averaged over the six UAs in China from 1999 to 2017
(**Fig. S6**).
**3.2. Associations of summer $O_3$ with WPSH**





301    Having established the relationships between summer $O_3$ and WPSH via the EOF

302  analysis, we further explore the direct responses of summer $O_3$ to WPSH. Since the

303  effects of the WPSH span vast regions, and the changes in surface ozone concentrations

304  may be influenced by the variations in meteorological factors associated with the

305  WPSH, a spatial correlation analysis between summer surface ozone concentrations in

306  China and WPSH (WPSH-I1) index from 1999 to 2017 was conducted. The result is

307  illustrated in **Fig. 4**. During this period, positive correlations overwhelm mainland

308  China, except for the PRD region (**Fig. 4**). Surprisingly, the negative correlations in the

309  PRD region might suggest that the stronger WPSH tends to reduce the summer $O_3$ in

310  this well-developed and populated UA in China, as aforementioned above. The summer

311  $O_3$ level in the PRD was the lowest among the six UAs (**Figs. 1** and **S3**). No statistically

312  significant $O_3$ trend was identified in the PRD, likely attributed to $O_3$ pollution control

313  in the early 2000s under the joint efforts from Hong Kong and Guangdong provincial

314  governments to improve the air quality in the PRD and Hong Kong (Wu et al., 2013).

315  We also estimated the correlations between $O_3$ concentrations averaged over the six

316  UAs across China and the WPSH-I1 from 1999 to 2017 (**Fig. S7**). The positive

317  correlation coefficients between the mean $O_3$ concentrations and the WPSH-I1 in each

318  of the UAs are presented at the top of each column**.** The results suggest that increasing

319  WPSH-I1 plays a specific role in elevated $O_3$ levels in eastern China and these UAs.

320  Again, as aforementioned, a decadal scale-increasing WPSH-I1 trend occurred since

321  the late 1990s and early 2000s (**Fig. S1**), which seems coincident with the rapidly

322  increasing $O_3$ precursor emissions and concentration trends in China and its major

323  urban areas (Liu and Wang, 2020; Lu et al., 2018). Hence, the positive correlations

324  between the summer WPSH-I1 and $O_3$ concentrations might not be attributable, to a

325  large extent, to growing $O_3$ pollution. Further discussions are presented in section 3.3.





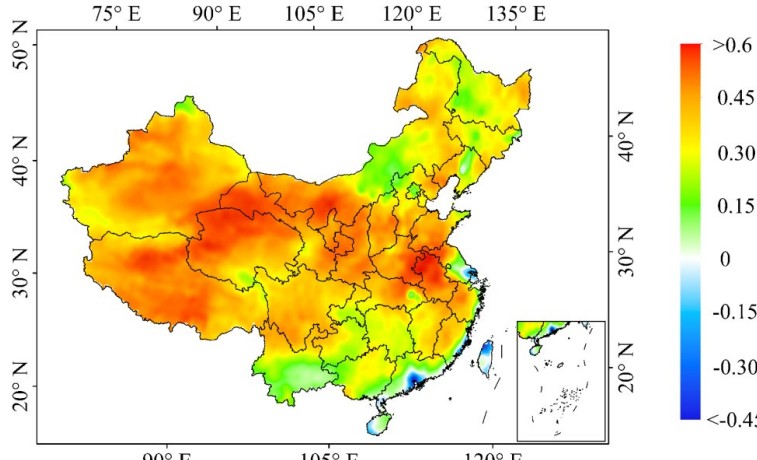

**Figure 4.** Correlation coefficients between summer O$_3$ concentrations and WPSH-I1 across China from 1999 to 2017 on the interdecadal scale.

Considering that summer precipitation in China is sensitive to the western ridge point of the WPSH, we also examined the responses of meteorological fields to the changes in the western ridge point index of the WPSH (referred to as the WPSH-I2) subject to its positive and negative phases. The WPSH-I2 is opposite to the WPSH-I1 (**Fig. S8**). We estimated WPSH-I2 anomalies as the departure from its mean from 1999 to 2017. We defined the positive WHSH-I2 phase if its values are greater than one standard deviation and the negative phase if WHSH-I2 < one standard deviation. The annual summer mean meteorological variables in the positive and negative phases of the WHSH-I2 are summed to obtain their respective composite means. We then calculated the anomalies of these composite means by subtracting their respective long-term means averaged from 1990 to 2022. **Figure 5** shows the anomalies of composite means of 500-hPa GH, precipitation (cm/mn), and the surface air temperature (SAT, ºC) across China in the positive and negative phases of WPSH-I2. The results identified evident north-south contrast for all three meteorological variables in the positive phase of the WPSH-I2. Of which, the anomalies of GH composite means are positive in northern China with a center in Mongolia and northeastern China (**Fig. 5a**). In contrast,



the broad region to the south of 35ºN is under the regime of negative GH composite
anomalies (**Fig. 5b**). The 500-hPa GH patterns can also be confirmed by the anomalies
of composite mean sea level pressure (SLP) in the positive and negative phase of
WPSH-I2 (**Fig. 6**), showing negative SLP anomalies from the Bay of Bengal to the
tropical western Pacific in the positive phase of the WPSH-I2 and positive anomalies
covering a vast region from southeast to northeast China. The south-north dipole
patterns of 500-hPa GH composite anomalies in **Fig. 5a** and the SLP composite
anomalies in **Fig. 6** often accompany the termination of the rain season in southern
China and the start of the rainy season in northern China (Nie et al., 2021), as shown
by the negative rainfall anomaly in southern China and positive anomaly in northern
China. **Figure 5a** predicts the weakening WPSH or the northward movement of the
WPSH, leading to a southward pressure gradient, as shown in **Fig. 5a**. As a result, the
composite anomalies of 850-hPa vector winds illustrate northerly wind components
over central-south and southern China (**Fig. 6**). Such northerly wind anomalies do not
favor southward water vapor transport by southwesterly Indian monsoon.

On the other hand, easterly and southeasterly wind components extend from

tropical west Pacific to central and northern China, paving a water vapor transport
pathway and corresponding to the positive rainfall anomaly in this part of China (**Fig.**
**5c**). In the negative phase of the WPSH-I2, positive SLP anomalies overwhelmed
eastern China with a center in the coastal region of southern China, implying the
enhancement of the WPSH. Accordingly, we observe negative composite anomalies of
the precipitation extending from the Yangtze-Huaihe Valley from central to
northeastern China, suggesting declining precipitations in these regions. Growing
precipitations are seen in southern and southeastern China, characterized by the positive
composite anomalies of the precipitation (**Fig. 5c**).

Precipitations in China have been connected strongly with the WPSH-I2 from a

daily perspective (Duan et al., 2008; Nie et al., 2021). Along with the westward shifting
of the WPSH ridge point, the major rain belt moves northward to the middle and lower
reaches of the Yangtze River from June to mid-July and northern and northeastern
China from late July to mid-August (Lu et al., 2017; Su et al., 2014; Zhao and Wang,



2017). In our case, with the focus on the association between summer $O_3$ and WPSH
from the interannual perspective, we show that the growing summer rainfall in southern
and southeastern China is associated with stronger WPSH in an east position, featured
by negative GH composite anomalies to the south of 35ºN in China (**Fig. 5a**). Such GH
anomaly pattern does not favor atmospheric water vapor transport to North China by
the summer monsoon circulations (Nie et al., 2021), which results in low rainfall in this
part of China (**Fig. 5c**). Accordingly, relatively higher SATs are observed in North
China (**Fig. 5e**), which, together with low atmospheric humidity and rainfall, favors $O_3$
formation and evolution. On the other hand, the stronger rainfall in southern and
southeastern China caused lower SATs in this region, characterized by negative SAT
composite anomalies (**Fig. 5e**). The higher atmospheric humidity, stronger rainfall or
precipitation washout, and lower SATs tend to restrain $O_3$ formation in southern China,
which resulted in lower $O_3$ levels compared to that measured in central and northern
China (Lu et al., 2018; Liu, 2020). This is likely a reason for higher $O_3$ concentrations
observed in northern and central-north China, such as the BTH and central China urban
agglomerations, than in YRD and PRD regions. In the negative phase of the WPSH-I2,
the north-south contrast pattern of all three meteorological variables vanished. Instead,
positive GH composite anomalies at the 500-hPa are seen in China, with more muscular
positive anomalies in western Mongolia (**Fig. 5b**). Such GH pattern suggests the
reinforcement and western shift of the WPSH. As a result, the composite anomalies of
the summer precipitation in northern China turned to positive, meaning high rainfall in
this region (**Fig. 5d**). However, the composite anomalies of SATs in the negative phase
of the WPSH-I2 (**Fig. 5f**) seem not to respond well to the intense rainfall, except in the
PRD, where declining SATs, featured by the negative SAT composite anomalies (**Fig.**
**5f**), corresponding well to the positive composite precipitation anomalies, meaning high
rainfall in this region (**Fig. 5d**). This result also is in line with previous observations
that the westward shift of the WPSH ridge point often accompanied with the
termination of the systematic rainfall in southern China (Duan et al., 2008; Huang et
al., 2018; Nie et al., 2021).






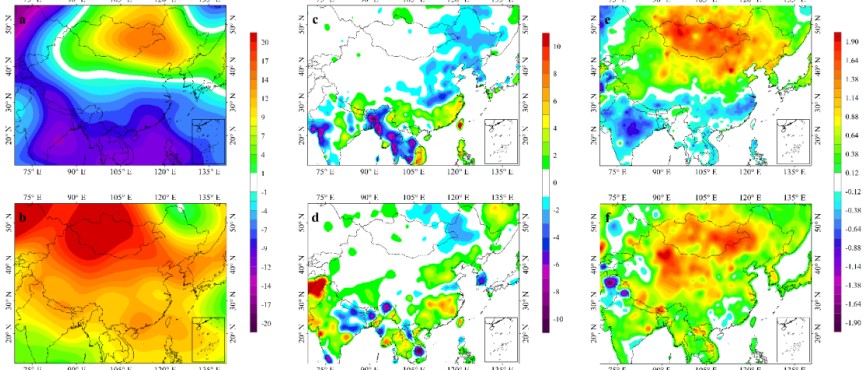


**Figure 5.** Anomalies of composite means of 500-hPa GH (GPH) in positive (**a**) and negative (**b**)
phase of WPSH-I2 from 1999 to 2017; same as Fig. 5a and 5b but for precipitation (cm/mn) in
positive (**c**) and negative (**d**) WPSH-I2 phase; same as Fig. 5a and 5b but for SAT (ºC) in positive
(**e**) and negative (**f**) phase of WPSH-I2 from 1999 to 2017.

413

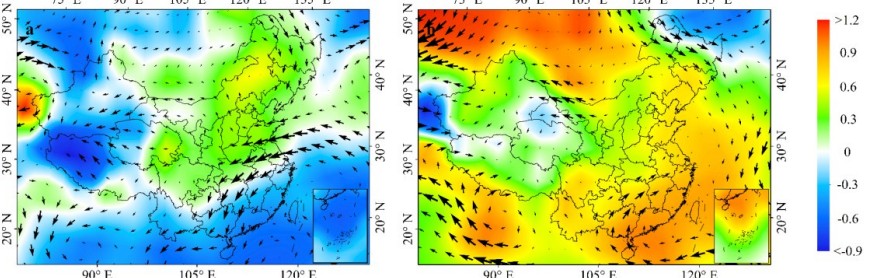

414

**Figure 6**. Anomalies of composite means of sea level pressure (SLP, hPa) overlapped with the
anomalies of composite mean 850-hPa vector winds across China in the positive (**a**) and negative
(**b**) phases of WPSH-I2 from 1999 to 2017.

418

**3.3. WPSH and interannual O₃ fluctuations**

Having identified the associations between the WPSH and O₃ evolution on
interannual scales, it is also interesting to know to what extent the WPSH could
contribute to the interannual fluctuations in O₃ concentrations in China and its major
urban agglomerations. We compared modeled O₃ concentrations among three model
scenarios by estimating their differences (fractions). **Figure 7** illustrates summer mean



O$_3$ concentrations averaged from 1999 to 2017 from the three scenarios. Identical
concentration spatial patterns can be observed in scenarios 1 (base, **Fig. 7a**) and 3 (fixed
meteorology, **Fig. 7c**), suggesting that precursor emissions overwhelmed the spatial-
temporal distribution of summer ozone in China. Comparing **Figs. 7b** with **Fig. 7a** and
**7c**, we also notice that the low summer ozone levels simulated from model scenario 2
(fixed precursor emissions, **Fig. 7b**) extend a much larger area across southern China
(highlighted by a solid red circle). Considering that model scenarios 1 and 2 used the
same meteorological data from 1998 to 2017, the lower O$_3$ levels under scenario 2 can
be attributed mainly to declining precursor emissions, partly attributable to a
collaborative effort to mitigate air pollution in the PRD and Hong Kong since the early
2000s as aforementioned before, which effectively slowed down growing O$_3$ precursor
emissions (Wu et al., 2013).

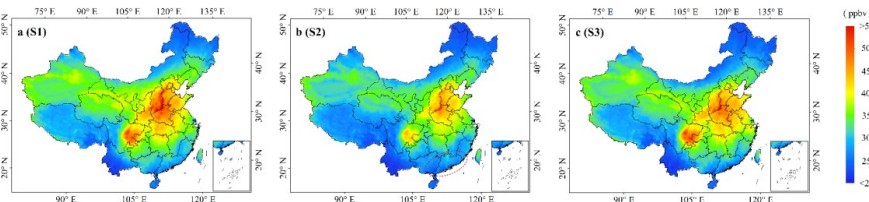

**Figure 7.** Modeled mean summer O$_3$ concentrations across China averaged from 1999 to 2017: (a)
Model scenario 1 (base scenario), (b) model scenario 2 (fixed precursor emission), and (c) model
scenario 3 (fixed meteorology).

To extract signals of meteorology in modeled O$_3$ concentrations, we calculated the
percentage change in summer O$_3$ concentrations subject to model scenarios 1 and 3,
defined as O$_{3,frac}$ = (O$_{3(S3)}$-O$_{3(S1)}$)/O$_{3(S1)}$ ×100%, where O$_{3(S3)}$ and O$_{3(S1)}$ represent the
summer ozone concentrations for scenarios 3 and 1 between 1999 and 2017 (**Fig. 8a**).
Since both model scenarios 1 and 3 used the same precursor emissions, their differences
(fractions) can quantify the meteorological effect on O$_3$ fluctuations. Significantly, the
WPSH was at a relatively high value in 1998 compared to 1999-2017 (**Fig. S2**). The
result shows that the fixed meteorological conditions (scenario 3) resulted in higher
summer ozone concentrations in the eastern seaboard region of China than the results





from the base scenario, particularly in the YRD, where the fixed meteorological
conditions enhanced the summer concentration by >9% compared to the base scenario
modeling result (**Fig. 8a**). The second-highest $O_3$ fraction between the scenarios 1 and
3 occurred in the Sichuan Basin, where the scenario 3 predicted the summer
concentrations are 3% to 6% higher than the base scenario 1. **Figure 8b** presents the
correlation coefficients between the WPSH-I1 and scenario 2 modeled $O_3$
concentrations across China from 1999 to 2017, showing relatively high positive
correlation coefficients in the eastern seaboard area and the region extending from the
Sichuan Basin to the Gansu-Shaanxi border, like the fractional changes shown in **Fig.**
**8a**. However, the negative correlations extended in most parts of China, indicating that
the WPSH tends to reduce summer $O_3$ levels in these regions. This spatial correlation
pattern differs significantly from the correlation pattern shown in **Fig. 4**, in which
positive correlations between the summer WPSH and modeled $O_3$ under the base
scenario almost extend entire China. As aforementioned, this is because both WPSH-
I1 and $O_3$ precursor emissions in China increased from 1999 to 2017. **Figure 8** shows
some similarities between spatial distribution patterns of the fractional changes in
summer $O_3$ concentrations under scenarios 1 and 3 and the correlations of summer $O_3$
concentrations from model scenario 2 and the WPSHI-I1. The result suggests that the
meteorological conditions contributing to summer $O_3$ evolution, as shown in **Fig. 8a**,
are associated, to a large extent, with the WPSH. The positive contribution of
meteorology characterized by the positive correlations to elevated $O_3$ pollution
gradually weakens in inland areas and turns into a negative contribution, meaning the
reduction of summer $O_3$ by meteorology in inland China, including most northern and
northeastern regions. Although positive correlations were estimated in the Tibet Plateau,
given very low $O_3$ pollution, the WPSH would not exert any significant influence on
$O_3$ levels in the plateau.





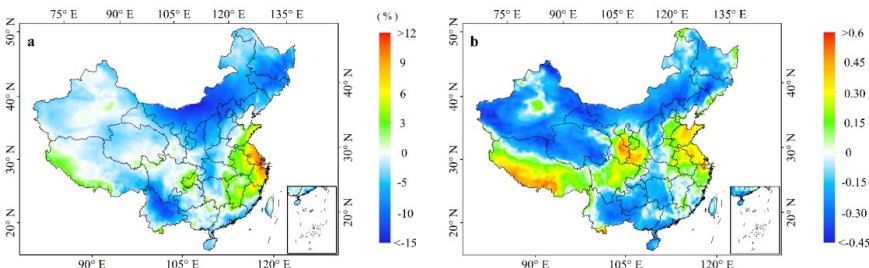

**Figure 8.** Fractional changes between modeled $O_3$ concentrations subject to model scenarios 3 and 1 from 1999 to 2017 estimated by $O_{3,frac} = (O_{3(S3)}-O_{3(S1)})/O_{3(S1)} \times 100\%$ (a), and correlation coefficients between summer WPSH-I1 and modeled summer $O_3$ concentrations under model scenario 2 (b).

As shown in **Fig. 5c,** the positive WPSH-I1 corresponds to lower precipitation in southern and southeastern China and higher precipitation in central and northern China, which tends to enhance $O_3$ levels in southern and southeastern China and reduce $O_3$ concentrations in the north. Although we also observe higher SATs across northern China and lower SATs in the south, which should increase $O_3$ levels in the north and decrease $O_3$ levels in the south, we could not quantify the direct linkages between SATs and $O_3$ concentrations from a national perspective. **Figure S9** displays the correlation coefficients between summer WPSH-I1 and the SAT (**Fig. S9a**) and precipitation (**Fig. S9b**). We can observe stronger positive correlations between the WPSH-I1 and SAT in southern China, indicating that the WPSH tends to enhance SAT in this region. This should favor elevated $O_3$ concentrations instead of the reduction of $O_3$ levels, as shown in **Fig. 7b**. This result likely anticipates that stronger precipitation associated with WPSH in this part of China overwhelmed SAT and overall yielded lower $O_3$ concentrations in southern China.

**Figure S10** illustrates annual variations of summer averaged $O_3$ concentrations under the three model scenarios from 1998 to 2017 over six UAs. Distinct differences between the three inland UAs (CY, CC, and MYR) and the three coastal UAs (YRD, PRD, and BTH) can be discerned from more significant fractions with an increasing trend in CY, CC, and MYR, as compared to YRD, PRD, and BTH, in which there were



no significant trends in modeled concentrations. In the three inland UAs (CY, CC, and
MYR, **Fig. S10d-f**), $O_3$ concentrations under fixed precursor emissions (scenario 2,
solid red line) are lower markedly than that from scenario 1(solid green line) and exhibit
no statistically significant temporal trend, suggesting that the variable meteorology
does not contribute significantly to $O_3$ levels and its long-term temporal trends. On the
other hand, $O_3$ concentrations under scenarios 1 and 3 runs are more or less similar and
illustrate increasing trends, indicating that growing precursor emissions in the past two
decades dominate long-term $O_3$ evolution in these inland UAs. In the three coastal UAs
(PRD, YRD, and BTH), the increasing trends of modeled summer $O_3$ under scenario 3
were less significant than in the three inland UAs, indicating slower growth of precursor
emissions in these coastal UAs. No significant increasing trends of $O_3$ concentrations
from scenario 3 run (fixed emission in 1998) are observed, suggesting that the changes
in meteorological conditions in the past two decades contributed less to growing $O_3$
pollution in China's major urban clusters than precursor emissions. However, we
noticed from **Fig. S10** that annual fluctuations of summer $O_3$ concentrations in these
UAs under scenario 2 agree, to a large extent, with the results from model scenario 1
(base scenario). This is expected because the two model scenarios shared the same
meteorology. As a result, precursor emissions contributed primarily to the long-term
$O_3$ growing trends and magnitudes, whereas meteorology made more vital
contributions to interannual fluctuations of $O_3$ concentrations.
To link the interannual fluctuations of summer ozone induced by meteorology
with WPSH, we estimated the rate of interannual variation (RIV) of summer $O_3$
concentrations simulated by scenario 2 in the 6 UAs and WPSH-I1, given by $C_r = [c(n)$
$– c(n-1)]/c(n-1) \times 100\%$, where $c(n)$ and $c(n-1)$ are summer $O_3$ concentrations in the
current year and previous year, respectively. The same approach also calculated the
RIV of summer WPSH-I1. **Figures 9a** and **9b** present the RIV of summer $O_3$
concentrations in the three inland and coastal UAs, respectively. The RIV of the WPSH-
I1 is also shown in the figure (brown dashed line). Although these RIVs do not exhibit
significant trends, we can observe a general agreement of the RIV between the
WPASH-I1 and summer $O_3$ concentrations in most UAs, featured by their annual



oscillations. **Figure 9c** shows the correlation coefficients between the RIVs of the
summer WPSH-I1 and $O_3$ concentrations. The highest correlation is found in the MYR,
followed by the PRD and YRD, whereas the lowest correlation occurred in the BTH.
These correlations suggest that the $O_3$ interannual fluctuations in those areas proximate
to the WPSH tend to be more strongly associated with the WPSH, regardless of the
positive or negative effect of the WPSH on $O_3$ evolution. Since the meteorology
determined largely the interannual fluctuations of summer $O_3$ and connected nicely with
the WPSH, the associations of the RIVs between summer the WPSH-I1 and $O_3$
concentrations imply that the WPSH made a more contribution to the interannual
variation of summer $O_3$, rather than its long-term trend, though the WPSH-I1 presents
an increasing trend after 1998 (**Fig. 3**).

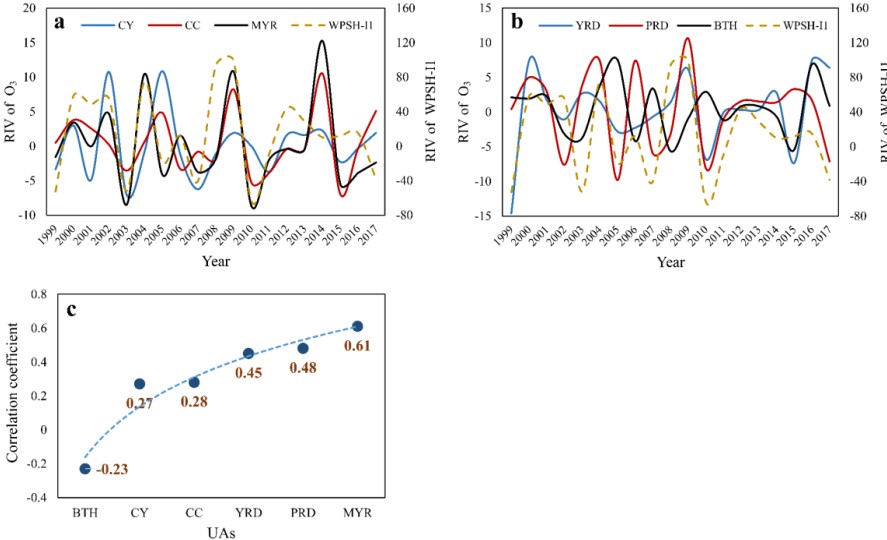

**Figure 9.** (a) Rate of interannual variations of summer WPSH-I1 (brown dashed line) and $O_3$ in CY,
CC, and MYR from 1999 to 2017, (b) same as Fig. 9a but for YRD, PRD, and BTH, and (c)
correlation coefficients of the rate of interannual variations of summer WPSH-I1 and $O_3$ in six UAs
from 1999 to 2017.

**4. Conclusions**



Model simulations revealed higher $O_3$ concentrations from 1999 to 2017 in the
Sichuan Basin and the region extending from central China to the NCP, agreeing with
measured mean summer concentrations. The first EOF loadings (PCA1) are associated
strongly with the mean summer $O_3$ concentrations across China and its major UAs. We
identified distinctive differences between positive and negative WPSH anomalies and
elucidated their impacts on interannual variation of $O_3$ and meteorological conditions.
In some of the UAs, such as the PRD, where relatively lower $O_3$ levels were reported
compared to other major UAs, the WPSH tended to reduce $O_3$ levels. The EOF and
regression analysis revealed stronger responses of summer $O_3$ in the region extending
from southeastern to central China. We noted that WPSH became stronger since the late
1990s and early 2000s, featured by the enhancing WPSH index after 1999. As a result,
stronger associations between summer $O_3$ in China and its primary UAs and the WPSH
occurred in the recent two decades. Extensive model scenario simulations indicated that
precursor emissions dominated the long-term trend and magnitude of summer ozone
concentrations. However, the meteorology associated with the WPSH largely
determined their interannual fluctuations from 1999 to 2017. Our results concluded that
the influence of precursor emissions on the evolution of ozone was stronger in
Chengdu-Chongqing, the middle reaches of the Yangtze River, and central China than
in the coastal city clusters. However, the influence of meteorological conditions is not
significant. In contrast, for the coastal city clusters of the Yangtze River Delta, the Pearl
River Delta, and the Beijing-Tianjin-Hebei region, the influence of precursor emissions
on the summer ozone evolution is weaker than in the inland city clusters, but the
influence of meteorological conditions was greater than in the inland city clusters,
particularly in those urban areas proximate to the WPSH. Considering the great efforts
in China to mitigate $O_3$ pollution via reducing anthropogenic precursor emissions,
interannual and longer-term $O_3$ evolutions associated with increasing WPSH strength
might be worth paying attention because it might influence background $O_3$
concentration, its long-term prediction, and long-term $O_3$ mitigation measures. The
results from the present study might also imply that the local policy makers in different
UAs should take the WPSH's impacts into account in making their respective $O_3$



reduction strategies, in addition to precursor emissions. To the end, it is worth noting
that this modeling study was partly based on an increasing trend of the summer WPSH
strength since 2000, which coincided with growing $O_3$ evolution. Historically, the
WPSH has been fluctuated on a yearly basis. Further study needs to be conducted to
discern the associations between projected WPSH and $O_3$ concentrations subject to
future climate change scenarios, such as shared socioeconomic pathways under
Coupled Model Intercomparison Project (CMIP6) and the Intergovernmental Panel on
Climate Change (O'Neill et al., 2017).

**Code/Data availability**

Data will be made available on request.

**Author contributions**

All authors contributed to the manuscript and have given approval of the final version.
XZ, RZ and XJ designed the research. XZ, XL and KC collected the data. ST, JL, HG
and TH contributed to the interpretation of results. XZ, RZ and JM wrote and revised
the manuscript.

**Competing interests**

The authors declare that they have no known competing financial interests or personal
relationships that could have appeared to influence the work reported in this paper.

**Financial support**

This study is supported by the National Natural Science Foundation of China
608 (41991312, 41977357).


**Appendix A. Supplementary data**

Supplementary data to this article can be found online

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
