# Peer review of "Western Pacific Subtropical High in China from 1999 to 2017"

_EGUsphere, 2023_

## Referee Comment (RC2)

The manuscript by Zhang et al. analyzes the influences of climate variations on long-term $O_3$ trends in China and explores the linkage between $O_3$ and a dominant atmospheric circulation system, using a modeled tropospheric ozone dataset and two western pacific subtropical high (WPSH) indexes. They conclude that the effect of the WPSH on regional $O_3$ is attributed to the changes in air temperature, precipitation, and winds associated with the WPSH's intensity and positions. However, the discussion of EOF analysis is lack of sufficient explanation on the association of $O_3$ patterns with WPSH. The significance of this paper is not expound sufficiently. The author need to highlight this paper's innovative contributions in abstract and conclusions. Here list some of my main concerns.

1. Climatologically, the WPSH activities with east–west expansion, and north–south movement significantly affect the daily, seasonal, interannual, and longer-term meteorological fields and climate variations over central and eastern China. which temporal scale of WPSH exerts the most significant effect on tropospheric $O_3$ in daily, seasonal, interannual, and longer-term variations? Please add more discussions on WPSH climatology and environment effects.

2. There are the distinct patterns in spatial distribution of WPSH with most significant seasonal (sub-seasonal) variations. Why can leads the WPSH to lower $O_3$ levels in the Pearl River Delta (PRD) region (line 32)? There is a misleading on the relation between the WPSH and lower O3 levels. How can WPSH affect the tropospheric $O_3$ over the Tibetan Plateau and Northwest China? It is suggested to focused the central and eastern China with the direct WPSH effect.

3. Lines 38-41: Please clarify how the effect of the WPSH on regional O3 depends on the spatial proximity to the WPSH. The WPSH position or spatial distribution is mostly controlled by the ridgeline of the WPSH with north–south shifts. why is the ridgeline index of WPSH not used in this study? The effects of the WPSH on $O_3$ interannual variations to the changes in air temperature, precipitation, and winds associated with the WPSH's intensity and positions. The tropospheric O3 is produced with photochemical reactions of O3 precursors under sunlight. How is the downward solar radiation as the most important factor of meteorology? Please

check the correlations.

4.   Lines 20-21:   The present study used a unique tropospheric O3 dataset. Please clarify how is the unique in the simulated dataset? Why are the WFR-Chem simulated meteorological elements not used the climatic analysis of atmospheric circulations?

5.   Text 1 & Fig S1: "Considering large uncertainties of sampled ambient air quality data in the first several years, we collected monitoring data in summer 2016 to verify modeled $O_3$ concentrations." Some stations were built in 2015, but the time period of sampled surface $O_3$ concentrations is still longer than one year in China. Why did author just choose the $O_3$ data in 2016 summer? The modeling results seems to be not very well in 2016, it is suggested to extend the observation dataset. Besides, due to the diurnal variation of $O_3$, the line chart is not the best way to present the reasonability of model simulation, makers without line would be better.

6.   Lines 152-153: "This trend possibly overwhelms interannual changes in the WPSH in the recent two decades." What does 'this trend' refer to? Growing $O_3$ pollution or strengthen WPSH?

7.   Fig S2: We cannot intuitively see the difference in the interannual trend of WPSH-I1 before and after 1999. Suggest to add the liner trend of WPSH-I1 from 1980 to 1999 in Fig. S2, to better display the reinforcement of the WPSH on a decadal scale in the recent two decades.

8.   Lines 172-174: "In the present study, we used the EOF analysis in WRF-Chem simulated gridded (20 km × 20 km) seasonal $O_3$ concentrations across China to extract annual O3 change features from 1999 to 2017, respectively." I am not quite clear on what 'respectively' refers to? EOF analysis for each year or at each grid?

9.   Line 243: "This inland region covers several major urban agglomerations (UAs) in China". 'UAs' has appeared in the previous context.

10. Lines 271-271: "Since $O_3$ concentrations are positively correlated with the WPSH-I1 (Figs. 3-5)" WPSH-I1 is not mentioned in Fig 5, please check the citation of figures.

11. Fig. 3: The relative analysis of the association of WPSH with PCA2 and PCA3 are

not yet described in the manuscript, please add them. Besides, third EOF pattern of $O_3$ is absent.

12. The time period for climate mean in Fig. S5 is 1999-2017, but it becomes 1980-2017 in Fig. S6. Why did author choose the different time periods for climate mean?

13. Fig.4 & Fig. 8: The correlation of observed surface $O_3$ concentration and WPSH-I1 is also significantly negative in YRD, which is not mentioned in the analysis of Fig. 4. However, the positive contribution of meteorology was characterized by positive correlation coefficients between the WPSH-I1 and scenario 2 modeled $O_3$ concentrations in the eastern seaboard area in Fig. 8b. The conclusions appear to contradict each other. Please provide an explanation.

14. Lines 315-316: "We also estimated the correlations between O3 concentrations averaged over the six UAs across China and the WPSH-I1 from 1999 to 2017 (Fig. S7). The positive correlation coefficients between the mean O3 concentrations and the WPSH-I1 in each of the UAs are presented at the top of each column." Fig. S7 is the correlation between $O_3$ concentrations and PCA1, please check the citation and add the legends. What do the Y-axis and X-axis of the inset figure stand for? Suggest to add the correlation coefficients in each subplot, which is more intuitive to illustrate the positive correlation than scatter plots.

15. Lines 332-333: "Considering that summer precipitation in China is sensitive to the western ridge point of the WPSH". It is necessary to cite some references.

---

## Referee Comment (RC3)

Review "Associations of interannual variation of Summer Tropospheric Ozone with Western Pacific Subtropical High in China from 1999 to 2017 " by Zhang et al.

General

Surface ozone can post great threats to public health and vegetation growth. Ozone pollution in China has become a severe environmental issue in the recent decades. Surface ozone varies at different time scale from diurnal to interannual scales. The interannual variation and long-term trend of surface ozone are difficult to investigate partially because of lack of long term observations. Therefore, numerical models become a powerful tool in addressing this issue. In this work, Zhang et al. used the Weather Research and Forecasting model coupled with Chemistry, WRF- Chem, to investigate interannual variations in summertime ozone for 18 years from 1999-2017 over China. Through EOF analysis and sensitivity simulation experiments, they linked summer ozone variation with the interannual variation in the Western Pacific Subtropical High (WPSH). The topic is suitable to Atmospheric Chemistry and Physics. The research ideas are innovative. The analysis are in some depth. The results are meaningful and interesting.

I provide the following comments/suggests for the authors to consider when revising their paper.

This is a simulation-based analysis. Therefore, how WRF-Chem performs is critical. The authors presented some validation validations at short time scales (Figure S1). How about at interannual scale? How well the model can capture the interannual variation and trend is most relevant to this work. The authors can use the recent (since 2013) surface measurement for this validation.

When the authors explored the underlying mechanisms for the linkage between summertime surface ozone and WPSH, they considered air temperature, precipitation, and wind (Abstract, Figures 5 and 6). Radiation is missing. As known, radiation is one of the most important drivers for surface ozone formation. Therefore, please take radiation into consideration.

There are many differences in the correlation of a WPSH index with surface ozone between Figure 4 and Figure 8a, which are puzzling. Can the authors please explain the differences?

One key figure seems missing: what are the spatial distributions of the composite anomalies of surface ozone in positive and negative phases of WPSH from the model simulations? How do the two distributions differ? The authors can compare these differences with those in recent observations (select two years with the largest difference in the WPSH index) and discuss your observations.

The authors can also briefly discuss relative importance of other climate modes, such as ENSO, and the East Asian monsoon to the interannual variation in surface ozone over China, comparing with WPSH.

Both abstract and conclusions lack of quantitative information (only two pieces of information in abstract, zero piece of information in conclusions). Please add more quantitative discussion.

Minor

Figure 1, please show the domain for the subregions studied (CY, CC, MYR, YRD, PRD, and BTH) in this figure or another figure.

Figures 4b and 8, please only show significant correlations, or indicate where the correlation is significant (p<0.05).

---

## Author Comment (AC1)

**Response to Reviewer #1's comments**

First of all, we would like to thank the Reviewer #1's comments and suggestions, which improved significantly the presentations and interpretations of our revised manuscript. In the revised article, we have addressed all comments from the Reviewer. Our point-by-point responses to the Reviewer's comments are outlined below. The original comments are shown in italics and our responses are given in normal fonts.

*This study utilized model simulations and data analysis to investigate the spatiotemporal variations of summer ozone concentrations in China and their influencing factors. The results revealed higher ozone concentrations in the Sichuan Basin and the central region of North China compared to other areas, and a strengthening correlation between summer ozone concentrations and the Western Pacific Subtropical High (WPSH) over the past two decades. Precursor emissions were identified as the dominant factor driving the long-term trends and magnitudes of summer ozone concentrations, while meteorological conditions associated with the WPSH played a key role in the interannual variability of ozone. The response of ozone evolution to precursor emissions and meteorological conditions varied across different urban areas, with inland city clusters exhibiting stronger responses to precursor emissions and coastal city clusters showing stronger responses to meteorological conditions. Therefore, the development of appropriate ozone reduction strategies should consider the specific characteristics and environmental conditions of each local urban area. Overall, I recommend the acceptance of the manuscript after making minor revisions.*

**Response:** We thank the Reviewer's positive and encouraging comments which help us improve this article considerably. We have made every effort to address the Reviewer's comments and questions.

**Point-by-point responses:**

1. *This study mainly analyzes the simulation results, but it seems there is a lack of model evaluation. To enhance the credibility of the paper, it is recommended to provide a comparison between the simulated results and measurements to validate the simulated ozone concentrations.*

**Response:** Following the Reviewer#1's recommendation, we have extended model result evaluation using 2016 sampling data to 2016 to 2017. Considering large uncertainties in measure $O_3$ concentrations due to artificial intervention, we did not implement sampled $O_3$ concentration measurements before 2016 in our model verification (Lines 130-133 in main text and SI Text 1 and Fig. S1). The results show better agreement between modeled and sampled $O_3$ concentrations of 2016 through 2017. Details were referred in the end of section 2.1 and revised SI Text 1 and Fig. S1.

*2. The first paragraph of the introduction describes various hazards of ozone, which, although accurate, are not closely related to the main topic of this paper. To quickly focus on the topic, it is advised to trim down these descriptions in the introduction and emphasize the background and objectives of the research.*

**Response:** Following the Reviewer's suggestion, we have revised Introduction section in which we cut down discussions on health risks of $O_3$ pollution, thereby enhancing the direct focus on objectives of this study.

*3. Two indices of the Western Pacific Subtropical High (WPSH) were employed in this study, but the description of the impact of the second index on ozone seems more like an inference and requires a more rigorous analysis.*

**Response:** Since summer rainfall in China was reported to be more sensitive to the western ridge point of the WPSH (Jiang et al., 2021; Yang et al., 2022; Zhao and Wang, 2017), which might affect the $O_3$ wet deposition, we also considered the westernmost point of the WPSH (hereafter referred to as WPSH-I2) in the present study. We found the strongest negative correlations between $O_3$ concentrations and the WPSH-I2, which is likely associated with $O_3$ washout by precipitation.

This point has been added to revised manuscript.

*4. The paper argues that the influence of WPSH on regional ozone depends on the spatial proximity to WPSH. Firstly, please clarify the geographic scope of WPSH. Secondly, this conclusion seems to be invalid in some regions, such as Xinjiang, where the correlation between ozone and WPSH is stronger than in Mongolia. To enhance the accuracy and applicability of the paper, please provide more detailed analysis and data support regarding the relationship between ozone and WPSH in different regions, and discuss possible reasons for these differences.*

**Response:** To address the Reviewer#1's comment, in the beginning of revised section 2.2, we added following statements "The WPSH is an anticyclonic system hovering over the middle and lower troposphere of the northwestern Pacific Ocean. The WPSH forms during the summer months and dissipates in winter. As a high-pressure system, the WPSH is associated with stable weather conditions featured by high temperature and low rainfall. These weather conditions, in turn, perturb significantly $O_3$ variation. While varying year from year, the WPSH in summer generally covers much of East Asia, including parts of China, Japan, and the Korean Peninsula. It can also extend westward, affecting Southeast Asia, including Vietnam, Thailand, and the Philippines (Jiang et al., 2021; Yang et al., 2022). Although the summer WPSH determines primarily the weather and climate conditions in Eastern and Southern China, it may also influence the weather systems in Western and Northern China. For example, the westward and northward movement of the WPSH might lead to a weak high-pressure system in Northern Xinjiang extending to Central-North China, resulting in higher temperatures and lower rainfall in this region, whereas a low-pressure system could prevail in Northern and Northeastern China, enhancing precipitation in this part of

China. However, given lower $O_3$ levels in Westernmost China (Tibet and Xinjiang), the present study did not attempt to elucidate the associations between $O_3$ evolution and the WPSH in this part of China but focused on Central and Eastern China where significantly higher $O_3$ levels were observed."

---

## Author Comment (AC2)

**Response to Reviewer #2's comments**

First of all, we would like to thank the Reviewer #2's comments and suggestions, which improved significantly the presentations and interpretations of our revised manuscript. In the revised article, we have addressed all comments from the Reviewer. Our point-by-point responses to the Reviewer's comments are outlined below. The original comments are shown in italics and our responses are given in normal fonts.

*The manuscript by Zhang et al. analyzes the influences of climate variations on long term $O_3$ trends in China and explores the linkage between $O_3$ and a dominant atmospheric circulation system, using a modeled tropospheric ozone dataset and two western pacific subtropical high (WPSH) indexes. They conclude that the effect of the WPSH on regional $O_3$ is attributed to the changes in air temperature, precipitation, and winds associated with the WPSH's intensity and positions. However, the discussion of EOF analysis is lack of sufficient explanation on the association of $O_3$ patterns with WPSH. The significance of this paper is not expound sufficiently. The author need to highlight this paper's innovative contributions in abstract and conclusions. Here list some of my main concerns.*

**Response:** We thank the Reviewer's positive and encouraging comments which help us improve this article considerably. We have made every effort to address the Reviewer's comments and highlight the innovative contributions of this paper in revised Abstract and Conclusions.

**Point-by-point responses:**

1. *Climatologically, the WPSH activities with east-west expansion, and north-south movement significantly affect the daily, seasonal, interannual, and longer-term meteorological fields and climate variations over central and eastern China. Which temporal scale of WPSH exerts the most significant effect on tropospheric $O_3$ in daily, seasonal, interannual, and longer term variations? Please add more discussions on WPSH climatology and environment effects.*

**Response:** To address the Reviewer's comment, we have added corresponding discussions on the impact of WPSH on daily and short-term $O_3$ variations from previous studies and potential causes in revised Introduction (the last paragraph). We did not attempt to identify the influences of WPSH with different temporal scales on tropospheric $O_3$ but focused on interannual and long-term scale effect because daily and short-term WPSH effects on $O_3$ have been investigated in China previously. Rather, the influences of WPSH on interannual and longer term $O_3$ variations are almost unknown, which was the major objective of our study. In revised Introduction (last paragraph), we further emphasized this objective.

Following the Reviewer's comment, we have added detailed descriptions on WPSH climatology and its effect on $O_3$ variation in the beginning of revised section 2.2.

2. *There are the distinct patterns in spatial distribution of WPSH with most significant seasonal (sub-seasonal) variations. Why can leads the WPSH to lower O₃ levels in the Pearl River Delta (PRD) region (line 32)? There is a misleading on the relation between the WPSH and lower O₃ levels. How can WPSH affect the tropospheric O₃ over the Tibetan Plateau and Northwest China? It is suggested to focus the central and eastern China with the direct WPSH effect.*

**Response:** We thank the Reviewer to indicate the potential misleading in our analysis. Following the Reviewer's comment, we have rewritten the first paragraph of section 3.2 by adding following statements "The causes of the lack of statistically significant O₃ trend and negative correlation between WPSH-I1 and O₃ in the PRD might be complex. The stronger WPSH and its westward extension can yield high temperature and dry weather condition in the PRD, which is conducive to elevated O₃ concentration, and vice versa. **Figure S7** shows relatively strong positive correlation between SAT and WPSH-I1, which favors growing O₃ concentrations, and negative correlation between precipitation and WPSH-I1 precipitation, which removes O₃ concentrations from air in the PRD region. From the early 2000s, Hong Kong and Guangdong provincial governments jointly lunched an O₃ pollution control program, which significantly reduced O₃ precursor emissions and its atmospheric levels in the PRD (Wu et al., 2013). It is likely that the course of O₃ reduction in the PRD coincided with the period of our modeling investigation, which interferes the statistical correlation between WPSH and O₃ in the PRD."

We agree with the Reviewer that the focus of this study should on Central and Eastern China. Considering that the WPSH is a most important summer weather and climate system in China, we briefly discussed its potential impact on weather conditions in Western and Northwestern China. In the revised first paragraph of section 2.2, we have added new statements "Although the summer WPSH determines primarily the weather and climate conditions in Eastern and Southern China, it may also influence the weather systems in Western and Northern China. For example, the westward and northward movement of the WPSH might lead to a weak high-pressure system in Northern Xinjiang extending to Central-North China, resulting in higher temperatures and lower rainfall in this region, whereas a low-pressure system could prevail in Northern and Northeastern China, enhancing precipitation in this part of China. However, given lower O₃ levels in Westernmost China (Tibet and Xinjiang), the present study did not attempt to elucidate the associations between O₃ evolution and the WPSH in this part of China but focused on Central and Eastern China where significantly higher O₃ levels were observed."

3. *Lines 38-41: Please clarify how the effect of the WPSH on regional O₃ depends on the spatial proximity to the WPSH. The WPSH position or spatial distribution is mostly controlled by the ridgeline of the WPSH with north-south shifts. why is the ridgeline index of WPSH not used in this study? The effects of the WPSH on O₃ interannual variations to the changes in air temperature, precipitation, and winds associated with the WPSH's intensity and positions. The tropospheric O₃ is*

*produced with photochemical reactions of O₃ precursors under sunlight. How is the down ward solar radiation as the most important factor of meteorology? Please check the correlations.*

**Response:** Firstly, because, as a large-scale high-pressure system, the WPSH affects significantly on its surrounding weather conditions, which, in turn, perturbs more strongly O₃ concentrations in its nearby regions. This point has been added to revised manuscript (lines 579-581).

Secondly, we did estimate correlations between seasonal O₃ time series and 4 WPSH indices, including the ridgeline index, results revealed low correlation compared with the area index and the western ridge point index. Please referred to rephrased 2ⁿᵈ paragraph of revised section 2.2.

Thirdly, following the Reviewer's suggestion, we have added a new Fig. S9 showing the correlation between O₃ concentrations and incoming (solar) radiation flux as well as the WPSH, and corresponding discussions in main text (lines 528-538).

4. *Lines 20-21: The present study used a unique tropospheric O₃ dataset. Please clarify how is the unique in the simulated dataset? Why the WRF-Chem simulated meteorological elements are not used the climatic analysis of atmospheric circulations?*

**Response:** "unique" dataset means the O₃ concentration dataset covering the longest time period because available O₃ time series data in China started from 2013 only. Nevertheless, we have deleted "unique" in the revised paper.

Yes, we used both WRF simulated meteorology and NCEP reanalysis data. Considering that WRF outputs forecasted meteorological data that might be subject to errors and uncertainties from different error sources in the model, whereas NCEP reanalysis provides objectively analyzed data based on observations, we selected the NCEP reanalysis in composite analysis. We have revised section 2.3 and added this point in the rephrased section.

5. *Text 1 & Fig S1: "Considering large uncertainties of sampled ambient air quality data in the first several years, we collected monitoring data in summer 2016 to verify modeled O₃ concentrations." Some stations were built in 2015, but the time period of sampled surface O₃ concentrations is still longer than one year in China. Why did author just choose the O₃ data in 2016 summer? The modeling results seems to be not very well in 2016, it is suggested to extend the observation dataset. Besides, due to the diurnal variation of O₃, the line chart is not the best way to present the reasonability of model simulation, makers without line would be better.*

**Response:** Thanks for the Reviewer's suggestions. The routine O₃ sampling started in 2013 in China but there were large uncertainties in measured data due to manual

intervention before 2016. In the revised paper, we have extended model evaluation from 2016 to 2016 to 2017 by adding on more year O₃ sampling data in 2017. Considering that present study focused on interannual and longer-term summer mean O₃ variation associated with the summer WPHI, we replaced hourly data by daily concentrations. Results reveal better agreement between modeled and measured concentrations, as refereed in revised SI Text 1 and Fig. S1. We still used line chart to illustrate the associations between modeled and measured O₃ time series. After replacing hourly data by daily time series, we can observe that modeled daily O₃ concentrations match well measurements in summer 2016 and 2017.

6. *Lines 152-153: "This trend possibly overwhelms interannual changes in the WPSH in the recent two decades." What does 'this trend' refer to? Growing O₃ pollution or strengthen WPSH?*

**Response:** It means WPSH trend. We have rephrased text.

7. *Fig S2: We cannot intuitively see the difference in the interannual trend of WPSH-I1 before and after 1999. Suggest to add the liner trend of WPSH-I1 from 1980 to 1999 in Fig. S2, to better display the reinforcement of the WPSH on a decadal scale in the recent two decades.*

**Response:** Thanks for the Reviewer's suggestion. We have added a trend line from 1980 to 1999 in new Fig. S2.

8. *Lines 172-174: "In the present study, we used the EOF analysis in WRF-Chem simulated gridded (20 km × 20 km) seasonal O₃ concentrations across China to extract annual O₃ change features from 1999 to 2017, respectively." I am not quite clear on what 'respectively' refers to? EOF analysis for each year or at each grid?*

**Response:** This was a typo error. We have deleted "respectively".

9. *Line 243: "This inland region covers several major urban agglomerations (UAs) in China". 'UAs' has appeared in the previous context.*

**Response:** The Reviewer is right! We have replaced urban agglomerations by UAs.

10. *Lines 271-271 "Since O3 concentrations are positively correlated with the WPSH-I1 (Figs. 3-5)" WPSH-I1 is not mentioned in Fig 5, please check the citation of figures.*

**Response:** Figs. 3-5 were changed to Figs. 3 and 4.

11. *Fig 3: The relative analysis of the association of WPSH with PCA2 and PCA3 are not yet described in the manuscript, please add them. Besides, third EOF pattern of O₃ is absent.*

**Response:** The analysis of PCA2 has been added. Since the third principal components (PCA3 and EOF3) were almost meaningless, they both are removed from the revised paper.

*12. The time period for climate mean in Fig S5 is 1999 2017, but it becomes 1980-2017 in Fig. S6. Why did author choose the different time period s for climate mean?*

**Response:** This was a typo error and has been corrected.

*13. Fig.4 & Fig. 8: The correlation of observed surface O₃ concentration and WPSH-I1 is also significantly negative in YRD, which is not mentioned in the analysis of Fig. 4. However, the positive contribution of meteorology was characterized by positive correlation coefficients between the WPSH-I1 and scenario 2 modeled O₃ concentrations in the eastern seaboard area in Fig. 8b. The conclusions appear to contradict each other. Please provide an explanation.*

**Response:** Likely we did not described clearly. **Figure 4** shows a negative correlation between modeled summer O₃ concentration and WPSH-I2 time series in the YRD under model scenario 1 but model scenario 2 yields a positive correlation (**Fig. 8b**). Since model scenario 1 took annually-altered O₃ precursor emissions into consideration, the negative correlation suggests that declining precursor emissions from 1999 to 2017 in the YRD overwhelmed the WPSH effect. After removed the effect of precursor emissions, the meteorology associated with the WPSH would help enhance O₃ concentrations in this region.

This argument has been added to revised manuscript.

*14. Lines 315-316: "We also estimated the correlations between O₃ concentrations averaged over the six UAs across China and the WPSH-I1 from 1999 to 2017 (Fig. S7). The positive correlation coefficients between the mean O₃ concentrations and the WPSH-I1 in each of the UAs are presented at the top of each column." Fig. S7 is the correlation between O₃ concentrations and PCA1, please check the citation and add the legends. What do the Y axis and X axis of the inset figure stand for? Suggest to add the correlation coefficients in each subplot, which is more intuitive to illustrate the positive correlation than scatter plots.*

**Response:** Given that PCA1 as the first principal component of summer O₃ time series is associated strongly with the mean summer O₃ concentrations averaged over the six UAs in China at the correlation coefficient of 0.95 (p<0.01) from 1999 to 2017, it seems not necessary to present the results as illustrated in Fig. S7. So we have deleted Fig. S7 in the previous version of the paper and corresponding discussion in main text (lines 315-325 of the original paper version).

*15. Lines 332-333: "Considering that summer precipitation in China is sensitive to the western ridge point of the WPSH". It is necessary to cite some references.*

**Response:** Following the Reviewer's suggestion, several new references are added and referred in the revised paper.

---

## Author Comment (AC3)

**Response to Reviewer #3's comments**

First of all, we would like to thank the Reviewer #3's comments and suggestions, which improved significantly the presentations and interpretations of our revised manuscript. In the revised article, we have addressed all comments from the Reviewer. Our point-by-point responses to the Reviewer's comments are outlined below. The original comments are shown in italics and our responses are given in normal fonts.

*Review "Associations of interannual variation of Summer Tropospheric Ozone with Western Pacific Subtropical High in China from 1999 to 2017" by Zhang et al.*

*General*

*Surface ozone can post great threats to public health and vegetation growth. Ozone pollution in China has become a severe environmental issue in the recent decades. Surface ozone varies at different time scale from diurnal to interannual scales. The interannual variation and long-term trend of surface ozone are difficult to investigate partially because of lack of long term observations. Therefore, numerical models become a powerful tool in addressing this issue. In this work, Zhang et al. used the Weather Research and Forecasting model coupled with Chemistry, WRF- Chem, to investigate interannual variations in summertime ozone for 18 years from 1999-2017 over China. Through EOF analysis and sensitivity simulation experiments, they linked summer ozone variation with the interannual variation in the Western Pacific Subtropical High (WPSH). The topic is suitable to Atmospheric Chemistry and Physics. The research ideas are innovative. The analysis are in some depth. The results are meaningful and interesting.*

*I provide the following comments/suggests for the authors to consider when revising their paper.*

**Response:** We thank the Reviewer's positive and encouraging comments which help us improve this article considerably. We have made every effort to address the Reviewer's comments and questions.

**Point-by-point responses:**

*This is a simulation-based analysis. Therefore, how WRF-Chem performs is critical. The authors presented some validation validations at short time scales (Figure S1). How about at interannual scale? How well the model can capture the interannual variation and trend is most relevant to this work. The authors can use the recent (since 2013) surface measurement for this validation.*

**Response:** Following the Reviewer's comment, in the revised paper, we have extended model evaluation from 2016 to 2016 to 2017 by adding on more year $O_3$ sampling data in 2017. The routine $O_3$ measurements started in 2013 in China but there were large uncertainties in measured data due to manual intervention before 2016. Considering

that present study focused on interannual and longer-term summer mean $O_3$ variation associated with the summer WPSH, we have replaced hourly data by daily concentrations. Results reveal better agreement between modeled and measured concentrations, as referred in revised SI Text 1 and Fig. S1.

*When the authors explored the underlying mechanisms for the linkage between summertime surface ozone and WPSH, they considered air temperature, precipitation, and wind (Abstract, Figures 5 and 6). Radiation is missing. As known, radiation is one of the most important drivers for surface ozone formation. Therefore, please take radiation into consideration.*

**Response:** We agree with the Reviewer! In the revised paper, we have added a new Fig. S9 in SI showing the correlation coefficients between surface incoming solar radiation flux and $O_3$ concentration and WPSH-I1 from 1999 to 2017 across China. We also added corresponding discussion in main text.

*There are many differences in the correlation of a WPSH index with surface ozone between Figure 4 and Figure 8a, which are puzzling. Can the authors please explain the differences?*

**Response:** Likely we did not described clearly. Figure 4 shows the correlations between modeled $O_3$ and WPSH under model scenario 1 with variable meteorology and precursor emissions, whereas Figure 8 illustrates correlations subject to model scenario 2 with fixed precursor emissions and variable meteorology. For example, Figure 4 shows a negative correlation between modeled summer $O_3$ concentration from model scenario 1 and WPSH-I2 time series in the YRD under model scenario 1 but model scenario 2 yields a positive correlation (Fig. 8b). Since model scenario 1 took annually-altered $O_3$ precursor emissions into consideration, the negative correlation suggests that declining precursor emissions from 1999 to 2017 in the YRD overwhelmed the WPSH effect. After removed the effect of precursor emissions in model scenario 2 subject to fixed precursor emissions, the meteorology associated with the WPSH would help enhance $O_3$ concentrations in this region. Therefore, the spatial distribution patterns of the two figures are significantly different.

This point has been added to the revised manuscript

*One key figure seems missing: what are the spatial distributions of the composite anomalies of surface ozone in positive and negative phases of WPSH from the model simulations? How do the two distributions differ? The authors can compare these differences with those in recent observations (select two years with the largest difference in the WPSH index) and discuss your observations.*

**Response:** We thank the Reviewer's advice. We did estimate the composite anomalies of surface $O_3$ but didn't present them in the paper. Considering that precursor emissions should dominate $O_3$ levels, it is not straightforward to identify signals from the $O_3$

response to WPSH during its positive and negative phase from precursor emissions. One way is to calculate the composite anomalies $O_3$ in positive and negative phase of the WPSH under model scenario 2 with fixed precursor emissions but the results cannot be compared to observations because measured $O_3$ concentrations are determined primarily by its precursor emissions.

*The authors can also briefly discuss relative importance of other climate modes, such as ENSO, and the East Asian monsoon to the interannual variation in surface ozone over China, comparing with WPSH.*

**Response:** In Introduction, we have discussed the relationships between $O_3$ and other interannual climate modes. Following the Reviewer's comment, we have rephrased text as "Using modeled summer $O_3$ time series across China from 1999 to 2017, we have examined the response of gridded summer $O_3$ concentrations to the East Asian Summer Monsoon Index (EASMI), Nino indices, and western North Pacific subtropical high index (WPSH-I), the three climate modes influencing significantly the summer weather and climate in China, on an annual basis in the six major UAs in China (Zhang et al., 2022). The correlation coefficients between the summer $O_3$ concentrations and the three climate modes from 1999 to 2017 are 0.54 (WPSH-I, p=0.016), 0.38 (Nino indices, p=0.105), and 0.27 (EASMI, p=0.267), respectively. The results revealed that interannual changes in summer $O_3$ averaged over these UAs were more significantly associated with the WPSH-I among three atmospheric teleconnection patterns. The finding motivates us to carry out more broad and deep investigations of the associations between the long-term change in summer $O_3$ and the WPSH, aiming to shed new light on the extent of the impact of climate variation on $O_3$ trends in urban China."

*Both abstract and conclusions lack of quantitative information (only two pieces of information in abstract, zero piece of information in conclusions). Please add more quantitative discussion.*

**Response:** Following the Reviewer's suggestion, we have revised Abstract and Conclusion sections and added more quantitative information.

*Minor*

*Figure 1, please show the domain for the subregions studied (CY, CC, MYR, YRD, PRD, and BTH) in this figure or another figure.*

**Response:** Done!

*Figures 4b and 8, please only show significant correlations, or indicate where the correlation is significant (p<0.05).*

**Response:** Following the Reviewer's advice, we have marked those significant correlations (p<0.05) in Figure 4 and 8b.